

# Assessing future shifts in habitat suitability and connectivity to old-growth forests to support the conservation of the endangered giant noctule

Mattia Iannella, Urbana Masciulli, Francesco Cerasoli,
Michele Di Musciano and Maurizio Biondi

Department of Life, Health & Environmental Sciences, Università degli Studi dell'Aquila,
L'Aquila, Italy

## ABSTRACT

**Background:** Suitable climate and availability of habitats for roosting, foraging, and dispersing are critical for the long-term persistence of bat species. The giant noctule (*Nyctalus lasiopterus*) represents one of the lesser-known European bats, especially regarding the environmental factors which shape its distribution.

**Methodology:** We integrated climate-based ecological niche models with information about topography and rivers' network to model weighted suitability for *N. lasiopterus* in the western Palearctic. The weighted suitability map was then used to estimate connectivity among the distinct occurrence localities of *N. lasiopterus*, as well as from these latter towards European old-growth forests, under current conditions and different combinations of future timeframes (2030, 2050, 2070) and shared socioeconomic pathways (SSPs 3.70 and 5.85).

**Results:** Current weighted suitability is highest in Andalusia, northern Iberia, southwestern France, peninsular Italy, coastal Balkans and Anatolia, with dispersed suitable patches elsewhere. A north-eastward shift of weighted suitability emerges in the considered future scenarios, especially under SSP 5.85. The major current ecological corridors for *N. lasiopterus* are predicted within a 'belt' connecting northern Spain and southwestern France, as well as in the Italian Alps. However, following changes in weighted suitability, connectivity would increase in central-eastern Europe in the future. The bioclimatic niche of the western *N. lasiopterus* populations does not overlap with those of the central and eastern ones, and it only overlaps with climatic conditions characterizing old-growth forests in western Europe.

**Conclusions:** The outcomes of our analyses would help in designing specific conservation measures for the distinct groups of giant noctule populations, favoring the possibility of range expansion and movement towards forested habitats.

Corresponding author
Francesco Cerasoli,
francesco.cerasoli@univaq.it

## INTRODUCTION

Climate is a critical limiting factor for most species (*Root, 1988*; *Buckley & Jetz, 2007*; *Sexton et al., 2009*; *Wiens, 2011*), determining the extent of suitable habitats as well as the possibility of range expansion at both regional (*Iannella, D'Alessandro & Biondi, 2018*; *Iannella et al., 2019*; *Jamwal et al., 2021*) and global (*Albouy et al., 2020*; *Barreto et al., 2021*) scales. In the context of increasingly rapid global warming (*IPCC, 2021*), climate influences on the broad-scale distribution of vertebrate species are predicted to possibly exceed the effects of land-use changes in the next decades (*Newbold, 2018*).

Bats (Chiroptera) represent one of the most diversified and widespread mammal groups (*Simmons, 2005*), and the possible impacts of climate change on their distribution, phenology, and population viability has recently gained attention (*Rebelo, Tarroso & Jones, 2010*; *Reusch et al., 2019*; *Di Gregorio, Iannella & Biondi, 2021*; *Haest et al., 2021*; *Smeraldo et al., 2021*). Indeed, their sensitivity to environmental stressors and the low reproductive rate of most species make these long-lived mammals particularly exposed to population declines as environmental conditions change (*Jones et al., 2009*; *Reusch et al., 2019*). Moreover, since bats play key ecological roles such as populations control on insects, dispersal of plant seeds and pollination (*Galindo-González, Guevara & Sosa, 2000*; *Medellín, Equihua & Amin, 2000*; *Russo & Jones, 2003*; *Cleveland et al., 2006*), they are being increasingly targeted in biodiversity conservation planning (*Bellamy, Scott & Altringham, 2013*; *Mendes et al., 2017*; *Ducci et al., 2019*).

The giant noctule, *Nyctalus lasiopterus* (Schreber, 1780), is the largest European bat species and one of the rarest (*Popa-Lisseanu et al., 2008*). Its currently fragmented distribution in the western Palearctic includes circum-Mediterranean, Balkans, and central/eastern European countries (*Alcaldé, Juste & Paunović, 2016*). Despite occurring in several countries, the giant noctule is nowadays listed as 'Vulnerable' within the IUCN Red List (*Alcaldé, Juste & Paunović, 2016*) under the A4c (*i.e.*, populations reduction) and C2a (i) (*i.e.*, fragmentation among several sub-populations) criteria. Indeed, *N. lasiopterus* is strongly dependent upon forest ecosystems as it uses hollows in broadleaved, and occasionally coniferous, mature trees for roosting (*Estok, Gombkötő & Cserkesz, 2007*; *Popa-Lisseanu et al., 2008*; *Crucitti, 2011*; *Alcaldé, Juste & Paunović, 2016*). Old-growth forests are particularly likely to represent fundamental habitats for this species as they usually host several cavities in standing dead trunks. The distribution of the known European old-growth forest stretches, as well as of the areas potentially hosting unmapped patches, is reported in *Sabatini et al. (2018)*: old-growth forests seem to cover about 0.25% of the European land territory, and they are mainly clustered in Fennoscandia, the Carpathians and the Balkans. Despite their invaluable importance in terms of hosted biodiversity and related ecosystem services, only about half of the known European old-growth forests are strictly protected (*Sabatini et al., 2018*). As deforestation affects most of the current range of *N. lasiopterus*, likely shrinking the size of its populations (*Alcaldé, Juste & Paunović, 2016*), the preservation of the remaining old-growth forests stretches may be critical for the long-term persistence of the giant noctule in Europe.

In this context, the ongoing global warming may pose additional threats to *N. lasiopterus* populations. On the one hand, increasing winter temperatures across the range of the giant noctule could promote the premature end of the hibernation phase, which in turn would determine higher energy demands during the activity period, possibly leading to higher mortality rates in case individuals are not able to retrieve sufficient trophic resources. On the other hand, changing climatic conditions could also indirectly menace this bat species by producing mismatches between its phenology and that of its prey. Indeed, *N. lasiopterus* feeds on insects during the warm season, but its diet shifts to nocturnally migrating passerine birds in autumn, probably to favour fat accumulation to sustain wintering (*Popa-Lisseanu, Bontadina & Ibáñez, 2009*); as climate change has been proven to contribute both to decreasing insects abundance (*Møller, 2020*) and to shifts in migration timings of passerine birds (*Sokolov, 2006*) in Europe, it represents a crucial factor to be considered with respect to the conservation of *N. lasiopterus*.

Notwithstanding, several studies at the local scale have investigated the ecology, spatial behaviour, and conservation status of *N. lasiopterus* populations (*Estok, Gombkötő & Cserkesz, 2007*; *Popa-Lisseanu et al., 2008*; *Fortuna et al., 2009*; *Estók, 2010*; *Naďo et al., 2019*) there is still a lack of comprehensive evidence about the possible effects of the accelerating climate change on its broad-scale distribution and inter-populations connectivity. In particular, the presence of ecological corridors allowing individuals to move across vast land swathes would favor gene flow among populations, an aspect which is poorly known so far except for a few populations in southern Spain which have been monitored for long time (*Santos et al., 2016*).

Here, we gathered up-to-date occurrence records of *N. lasiopterus* across its full European range and implemented Ecological Niche Models (ENMs) *sensu Peterson & Soberón (2012)* to investigate climate influences on its potential distribution for the current timeframe as well as for various future scenarios. Moreover, given the above-mentioned importance of old-growth forests to the giant noctule, we processed the output from the fitted ENMs using ecological corridors modelling techniques based on circuit theory (*McRae et al., 2008*). In this manner, we assessed present and future connectivity (and the corresponding variations) among *N. lasiopterus* current populations, as well as between these latter and the remaining European old-growth forests. We then used evidence from this set of analyses to provide novel insights about the actual conservation status of the giant noctule, along with suggestions on how to manage, and possibly extend, the residual ecological corridors between *N. lasiopterus* populations and European old-growth forests.

## MATERIALS AND METHODS

### Study area and spatial data

We selected the western part of the Palearctic region (longitude range: $\sim -10° \div 60°$ E) as the study area. This choice was determined by combining information about *Nyctalus lasiopterus* current range, as assessed within the IUCN Red List of Threatened Species (*Alcaldé, Juste & Paunović, 2016*), with occurrence data from published literature and online resources.

In detail, the species' range spans from Portugal, Spain, France, Italy, Greece, Cyprus, and Turkey (with few scattered historical records also in northern Morocco and eastern Libya), to various countries in the Balkans (Slovenia, Croatia, Albania), central and eastern Europe (Slovakia, Poland, Belarus, Hungary, Bulgaria, Romania, Ukraine, European Russia), also extending to the Caucasus (Georgia, Azerbaijan) and western Kazakhstan (*Alcaldé, Juste & Paunović, 2016*).

The species' occurrence localities were gathered through literature search and the Global Biodiversity Information Facility (*GBIF, 2022*). As for the literature data, we excluded old records (before the 1970s) and those lacking precise geographic information about the occurrence locality (*e.g.*, articles generally reporting the presence of the species within a Region/Province or an entire Protected Area). Moreover, we excluded GBIF records being duplicates or reporting geographic coordinates with a spatial resolution coarser than 2.5 arc-min (*i.e.*, ~5 km). These filters permitted to select *N. lasyopterus* occurrence records with a spatial and temporal resolution matching that of the predictors used to fit the ENMs (*Sillero & Barbosa, 2020*).

To assess the climate-occurrence relationships of *N. lasiopterus*, we downloaded a set of 19 raster layers representing bioclimatic variables from the Worldclim 2.1 (*Fick & Hijmans, 2017*) archive (https://www.worldclim.org/data/index.html, accessed on 9 March 2022), at 2.5 arc-min resolution for the 'current' timeframe (*i.e.*, 1970–2000 average climatic conditions) as well as for three future time horizons (*i.e.*, 2030, 2050, and 2070). For each future horizon, we downloaded raster data representing predicted climatic conditions under two Shared Socioeconomic Pathways (SSPs). Specifically, we selected the SSPs 3.70 and 5.85: the latter predicts an increase in Earth's radiative forcing of 8.5 W m$^{-2}$ by 2100, due to lack of international effort in limiting greenhouse gas (GHG) emissions, and is frequently named 'business as usual' scenario (*Riahi et al., 2017*); differently, SSP 3.70 is usually labelled as 'middle of the road' because it predicts an increase in radiative forcing up to 7.0 W m$^{-2}$ in the next decades, in a context of "regional rivalry" with limited global cooperation to mitigate anthropogenic global warming (*Riahi et al., 2017*).

Predictions from the climate-based ENMs were then refined in a post-modelling phase (see below) by including topographic and habitat-related predictors. As for topography, we downloaded a Digital Elevation Model (DEM, 25 m resolution) from the European project Copernicus repository (https://land.copernicus.eu/imagery-in-situ/eu-dem, accessed on 9 March 2022). The habitat-related variables were instead derived from two distinct datasets: (i) vector spatial data about European rivers from *Grill et al. (2019)*, from which we selected the major riverine areas (*i.e.*, RIV_ORD > 4 based on the Strahler stream order, indicating the level of branching of watercourses in a top-down fashion) as these latter are used by *N. lasiopterus* to roost, forage, and move across vast landscapes (*Popa-Lisseanu, Bontadina & Ibáñez, 2009*); (ii) occurrence localities of European old-growth forests from *Sabatini et al. (2018)*. Occurrence localities of old-growth forests were converted from point to polygon features through a three-step procedure: first, we derived the radius (in meters) of the considered forest from the corresponding area, reported in hectares in *Sabatini et al. (2018)*, through the formula $\sqrt{(old\ growth\ forest\ area * 10000)/\pi}$; then, we used this radius to calculate a buffer around the occurrence point of the forest; finally, we
refined the obtained polygon by discarding the areas which were encompassed by the buffer but did not fall into forests' features of the European Nature Information System (EUNIS) spatial dataset (100 m resolution), which we downloaded from the European Environment Agency website (https://www.eea.europa.eu/data-and-maps/data/, accessed on 9 March 2022). Details about each of the considered old-growth forests, including the extent of the corresponding refined polygonal feature, is reported in Table S1.

## Ecological niche modelling

The ecological niche modelling step was performed in R version 4.1.1 (*R Core Team, 2021*). Specifically, we took advantage of the 'gbm' R package ver. 2.1.8 (*Greenwell, Boehmke & Cunningham, 2019*) to fit the ENMs for *N. lasiopterus* based on the gradient boosting model (GBM) algorithm, known also as boosted regression trees (*Elith & Graham, 2009*). The choice of this algorithm derived from both the possibility of easily tuning it in the R environment and to the fact that it emerged as one of the best performing ENM algorithms with presence-pseudo-absence data, once properly tuned (*Elith & Graham, 2009*; *Hao et al., 2020*).

To feed the GBM algorithm, we selected a subset of the 19 Worldclim bioclimatic variables by integrating the results of a Variance Inflation Factor analysis (VIF), performed through the 'usdm' R package (*Naimi, 2015*), with relevant published information about the species' ecology (*Cerasoli et al., 2021*). We chose VIF ≥ 10 as exclusion criterium for the single variables as it is deemed a suitable threshold to deal with multicollinearity in Ecological Niche Modelling (*Guisan, Thuiller & Zimmermann, 2017*). Then, we generated 5,000 pseudo-absences through the 'disk' strategy of the 'BIOMOD_FormatingData' function of the 'biomod2' R package (*Thuiller, Georges & Engler, 2016*), setting 130 and 250 km as the minimum and maximum radius, respectively. The rationale of this choice was threefold. First, selecting pseudo-absences within a geographically constrained buffer permits to avoid them falling either too close to the presence points, which could hamper a proper model calibration, or too far from them, which could artificially inflate model performance estimates (*VanDerWal et al., 2009*). Secondly, GBM was shown to perform well when pseudo-absences are located one or two degrees away (*i.e.*, roughly 120-to-240 km within the latitudinal range of our study area) from presence points (*Barbet-Massin et al., 2012*). Lastly, including ecological knowledge about the dispersal capability of the target species usually decreases the risk of pseudo-absences being instead false absences (*Phillips et al., 2009*): as the maximum daily foraging transit recorded for *N. lasiopterus* is about 130 km (*Naďo et al., 2019*), selecting pseudo-absences outside this minimum distance from presence points reduces the risk of labelling as absences localities potentially used by the giant noctule populations comprised in our occurrence dataset.

Then, we weighted presences and pseudo-absences so that the sum of the weights of the former equals the one of the latter. Indeed, assigning a same overall weight to presences and pseudo-absences usually increases ENMs' predictive performance when the generated pseudo-absences are far more numerous than the available presences (*Cerasoli et al., 2017*; *Gouvêa et al., 2020*; *Thiault et al., 2020*).

To detect the best parametrization for the GBM algorithm, we built three different matrices containing several combinations of 'gbm' parameters and the respective set of values (for brevity, here we list the ones for the first matrix only: *shrinkage* = 0.01, 0.1, 0.3; *interaction.depth* = 1, 3, 5; *n.minobsinnode* = 5, 10, 15; *bag.fraction* = 0.65, 0.8, 1). Then, we ran as many GBM models as the combinations, increasing the *n.trees* value from 10,000 to 20,000 but keeping the *train.fraction* = 0.8 and the *cv.folds* = 10 fixed. We finally chose the set of parameters resulting in the lowest root mean square error (RMSE) (*Friedman, 2001*; *Greenwell, Boehmke & Cunningham, 2019*; *Cervellini et al., 2021*). To investigate possible spatial autocorrelation in predictions from the optimized GBM model, which could have resulted for instance from sampling bias affecting *N. lasiopterus* occurrence data (*Roberts et al., 2017*), we fitted an additional set of 10 GBM models, using the tuned set of parameters. In each of the 10 iterations, the model was fitted on the occurrence data coupled with a set of 5,000 pseudo-absences randomly selected within the above-mentioned 130-to-250 km buffer. Then, we computed residuals on the calibration data for each of these models. Subsequently, we draw a correlogram, through the 'ncf' R package version 1.2-9 (*Bjornstad, 2020*), showing the variation of spatial autocorrelation, as represented through the Moran's index (I), at increasing average inter-point distances. Finally, we computed the mean and median pairwise Euclidean distances across *N. lasiopterus* occurrence data to check whether they fall within the inter-point distance range not affected by spatial autocorrelation in model predictions, as emerging from the correlogram.

Successively, we assessed the discrimination power of the optimized GBM model through the Boyce index (*Boyce et al., 2002*), which is particularly suited for ENMs built on presence and pseudo-absence data (*Hirzel et al., 2006*; *Leroy et al., 2018*). Moreover, we measured the relative contribution of the selected variables through the randomization algorithm implemented in the 'summary.gbm' function of the 'gbm' R package.

Then, we projected the optimized GBM model across the entire study area for both current climatic conditions and various future scenarios represented by the combinations of year (2030, 2050, and 2070) and SSP (SSP3.70 and SSP5.85). We buffered possible differences in future predictions, which may be observed when comparing ENMs' projections obtained using distinct General Circulation Models (*Stralberg et al., 2015*), by considering three of them, namely the BCC-CSM2-MR (*Wu et al., 2019*), the IPSL-CM6A-LR (*Boucher et al., 2020*) and the MIROC6 (*Tatebe et al., 2019*). Moreover, we first assessed the prediction uncertainty due to extrapolation (*i.e.*, the divergence between environmental conditions at calibration points and those across the projection surface) through the Multivariate Environmental Similarity Surface (MESS) (*Elith & Leathwick, 2009*) implemented in the 'dismo' R package (*Hijmans & Elith, 2016*). Then, we used the resulting MESS maps to implement the Multivariate Environmental Dissimilarity Index (MEDI). This index weighs ENMs' projections under different GCMs based on the corresponding MESS, finally returning a combined weighted projection (*Iannella, Cerasoli & Biondi, 2017*). We iterated this process of ENMs' refinement for each year × SSP combination. ENMs' predicted suitability, ranging from 0 to 1 (low-to-high suitability), was then reclassified on a 1-to-10 scale for post-modelling purposes.

## Post-modelling analyses

To sharpen the ENMs-derived predicted suitability for *N. lasiopterus*, we applied the "couple-and-weigh" framework from *Iannella et al. (2021a)* by integrating spatial information about elevation and rivers' network. All the post-modelling processes subsequently described were performed in ArcGIS Pro 2.9 (*ESRI Inc, 2022*).

First, we extracted elevation values at occurrence localities from the DEM so as to draw an elevation preference curve, normalizing the "raw" occurrence frequencies along the elevation gradient (100-m-wide intervals) to a 1-to-10 scale. Then, we took advantage of the information about "Connectivity Status Index (CSI)" and "Urbanization (URB)" present in the rivers' dataset of *Grill et al. (2019)* to calculate a connectivity-functionality index (obtained as $CSI-URB$), in turn converted to a 1-to-10 scale. In this manner, spatial data about elevation and rivers' network had the same scale of values (*i.e.*, 1-to-10) as the reclassified predicted suitability from the climate-based ENM.

Finally, to obtain current and future weighted suitability maps, we combined the current and future (MEDI-corrected) predictions from the climate-based ENMs with the reclassified DEM and rivers layers through the 'weighted overlay'. In this step, the cell resolution of the elevation and rivers layers was automatically upscaled to that of the ENM predictions (*i.e.*, 2.5 arc-min) by the 'weighted overlay' tool itself.

## Landscape connectivity

Starting from the obtained current and future weighted suitability maps, we calculated connectivity across the study area through the Circuitscape v.5 (*Anantharaman et al., 2020*) package in Julia programming language (*Bezanson et al., 2017*). Circuitscape, which was shown to perform well in ecological connectivity studies (*Dickson et al., 2019*), applies coupled random walk and circuit theory-based algorithms to model ecological connectivity across landscapes using a suitability (or friction) map as a base, and target/destination points (or areas) to be connected as nodes (*McRae et al., 2008*; *McRae, Shah & Edelman, 2016*). Two main node categories were used, with subsequent sub-algorithms applied, as we assessed both connectivity among *N. lasiopterus* occurrence localities and connectivity from these latter towards European old-growth forests. For the former aim, we applied the 'pairwise' approach so that, for each possible pair of occurrence localities, each term is, iteratively, the source or the destination node (*McRae, Shah & Mohapatra, 2013*). For the latter aim, we chose the 'Advanced' mode to select *N. lasiopterus* current populations as sources and old-growth forests as destination areas (*McRae, Shah & Mohapatra, 2013*). Then, we calculated, for both the pairwise and the directional (*i.e.*, occurrences-to-forests) case, the Standardized Connectivity Change Index (SCCI) between current and future scenarios, which ranges from −1 (connectivity loss) to +1 (gain), with 0 representing the connectivity stability (*Iannella et al., 2021a*, *2021b*).

Considering the daily distances that the giant noctule may cover (*Naďo et al., 2019*), the connectivity estimates were refined by applying a 130 km buffer around old-growth forests to highlight those falling within this distance from *N. lasiopterus* occurrence localities (*i.e.*, the forests being easily reachable by the species).

### Bioclimatic niche overlap and gap analysis

To assess possible climatic links between *N. lasiopterus* populations and old-growth forests, we used the 'hyperoverlap' R package (*Brown, Holland & Jordan, 2020*). Specifically, we segregated occurrence localities of *N. lasiopterus* and old-growth forests in three "longitude groups" (Longitude ranges: −15°–0°, Western; 0°–15°, Central; 15°–50° Eastern). Then, we first assessed overlap in bioclimatic space among the three groups of *N. lasiopterus* occurrences (42 Support Vectors, SVM kernel = linear), by adapting the 'hyperoverlap_set' function (*Brown, Holland & Jordan, 2020*) to perform the analysis with more than two groups (modified function reported in Supplemental Material). Successively, we assessed niche overlap between each longitude group of *N. lasiopterus* occurrences and each category of old-growth forests (*e.g.*, 'Near-virgin Forest', 'Long Untouched Forest'), divided by longitude group as well (104 Support Vectors, SVM kernel = polynomial, 2nd degree).

Also, for each longitude group we measured: (i) the distance between the *N. lasiopterus* occurrence localities and the old-growth forests; (ii) the values of the most contributing bioclimatic variables (as resulting from ENMs) within the occurrences of both *N. lasiopterus* and old-growth forests.

We finally evaluated the protection status of *N. lasiopterus* by performing a gap analysis (*Scott et al., 1993*; *Jennings, 2000*; *Hermoso et al., 2022*). Specifically, we assessed the number of occurrence localities falling within protected areas, as well as their distance from the easily reachable old-growth forests (*i.e.*, those located within the 130 km buffer). For this purpose, we downloaded spatial data about the Natura 2000 sites and Nationally Designated Protected Areas (https://ec.europa.eu/environment/nature/natura2000/data/index_en.htm and https://www.protectedplanet.net/en/thematic-areas/wdpa?tab=WDPA, respectively; accessed on 9 March 2022).

## RESULTS

### Occurrence data and climate-based ecological niche models

At the end of the filtering phase, we selected 260 occurrence localities for *Nyctalus lasiopterus* (Fig. 1, Table S2).

Average inter-point distance within the filtered occurrence dataset equaled 1,562 km, while the median inter-point distance was 1,349 km. The correlogram drawn on residuals from the 10 additional runs of GBM fitting indicated that spatial autocorrelation was practically absent (*i.e.*, Moran's I ~ 0) when inter-point mean distance ranged between 1,250 and 2,000 km (Fig. S1). Thus, we may be quite confident that no spatial bias possibly hidden in the occurrence dataset noticeably affected the ENM fitting process and the resulting predictions.

Based on the VIF scores (Table S3), the climatic variables selected for model fitting were BIO1 (annual mean temperature), BIO2 (mean diurnal temperature range), BIO6 (minimum temperature of the coldest month), BIO12 (annual precipitation), and BIO17 (precipitation of the driest quarter). The lowest RMSE was recorded for the GBM model fitted with 15,380 trees, *interaction.depth* = 3, shrinkage = 0.001 and *n.minobsinnode* = 15,
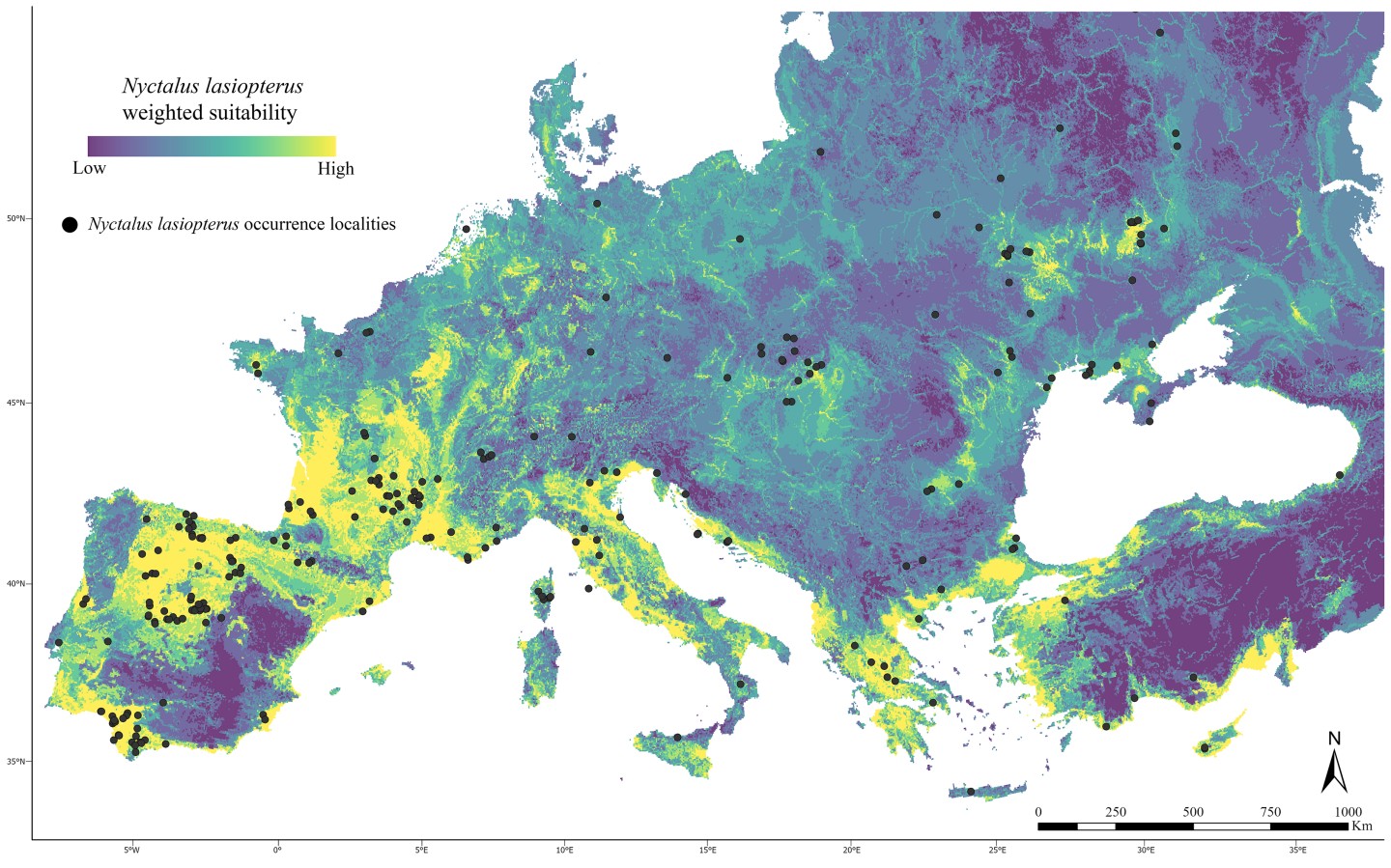

**Figure 1 Current weighted suitability map.** Predicted weighted suitability for *Nyctalus lasiopterus* (occurrence localities indicated as black dots) across the study area under current climatic conditions.

which also attained Boyce index = 0.867. The three most contributing variables within this model were: BIO6 (38.2%), which showed a sigmoid-shaped curve of predicted suitability peaking at ~−1 °C; BIO1 (18.3%), with suitability linearly decreasing after ~0 °C and BIO2 (17.5%), showing a peak in the 6–12 °C interval (marginal response curves are reported in Figs. S2A–S2C). Also, BIO6 – BIO2 represented the pair of variables attaining the highest interaction score (0.385), with highest predicted suitability when BIO6 ≥ 5 °C and BIO2 ranges between 8 °C and 12 °C (Fig. S1D).

## Weighted model

After the implementation of the "couple-and-weigh" step, in which the reclassified map from the optimized climate-based ENM was merged with the reclassified information about elevation and major rivers, the resulting weighted model predicted highly suitable areas (*i.e.*, weighted suitability >0.8) within an extent spanning −9° ÷ 35° in longitude and 36° ÷ 47° in latitude (Fig. 1), and scored a Boyce index = 0.903.

Under current conditions, areas in the southern and northern Iberian Peninsula, southwestern France, peninsular Italy, coastal areas of the Balkans, and coastal Anatolia

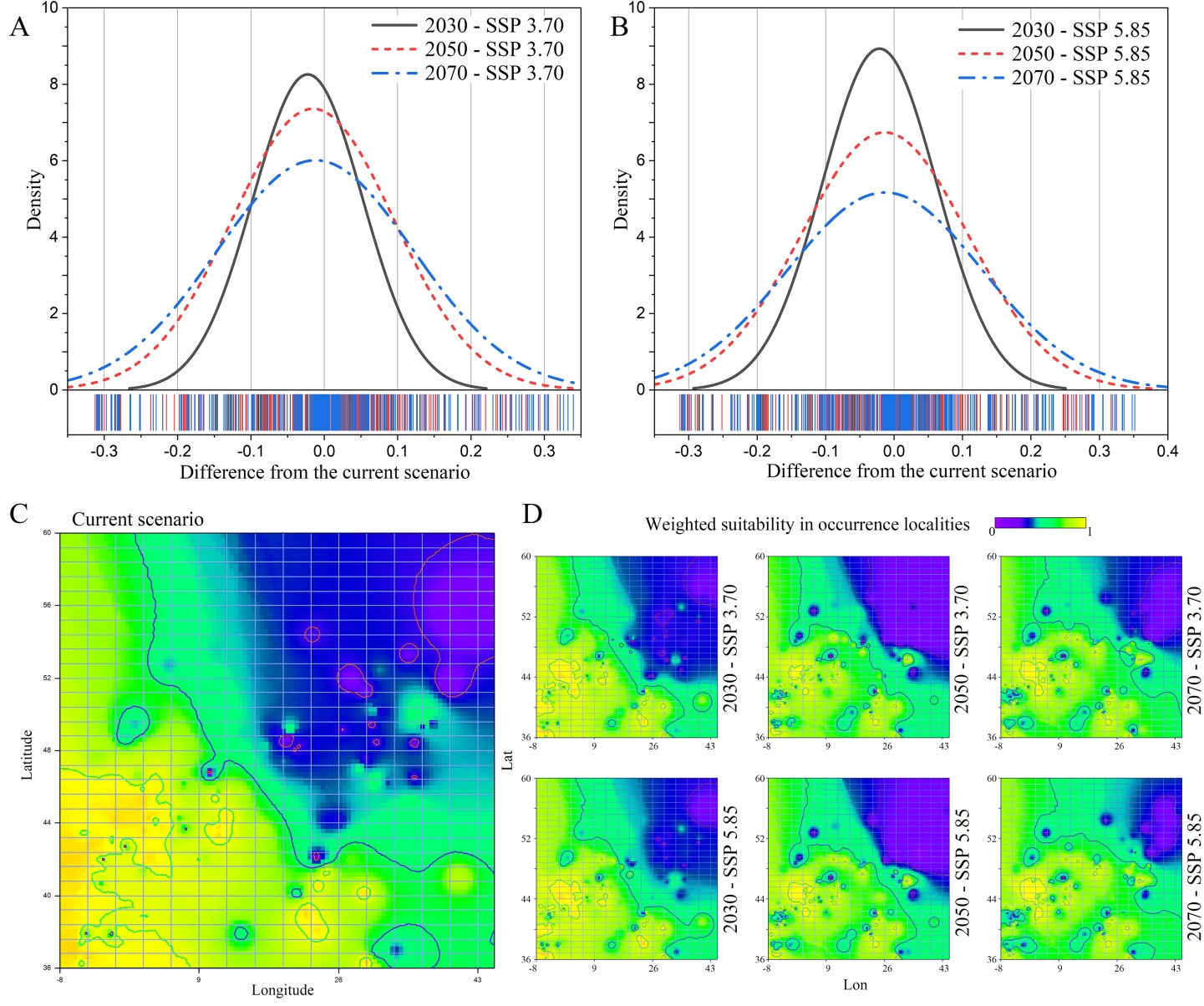

**Figure 2 Current-to-future suitability shifts.** Distribution of difference in weighted suitability for *Nyctalus lasiopterus*, sampled within the gathered occurrence localities, between current conditions and each of the three considered future time horizons (2030, 2050, 2070) under the Shared Socioeconomic Pathways (SSPs) 3.70 (A) and 5.85 (B). Interpolation of *N. lasiopterus* weighted suitability, measured within occurrence localities, across the extent of the study area under current conditions (C) and the considered future scenarios (D).

host the highest weighted suitability (Fig. 1). On the contrary, only small suitable patches are predicted in central and eastern Europe, corresponding to occurrence clusters.

A general decrease in weighted suitability on the gathered *N. lasiopterus* occurrence localities emerged from current to future scenarios, with suitability losses being more prominent as the time horizon of model projections moves forward (Figs. 2A and 2B). In 2030, the percent suitability loss compared to current conditions is essentially equal under the SSP 3.70 (−8.05%) and SSP 5.85 (−7.61%) scenarios. In 2050 and 2070, predicted

losses are even larger, bringing the total reduction to −17.90% and −17.31%, for the SSP 3.70 and SSP 5.85 scenarios, respectively.

The western areas, in particular those in the range of longitude −8° ÷ 30° and latitude 36° ÷ 48°, will experience the broadest suitability losses, with stronger evidence under the SSP 5.85 scenario (Figs. 2C and 2D). Our models, on the other hand, predict a significant increase in suitability in the north-eastern areas, particularly in the range of longitude 26° ÷ 45° and latitude 44° ÷ 60°, especially under the SSP 5.85 scenario, most likely as a result of possible temperature increases in the northernmost latitudes (Figs. 2C and 2D).

## Landscape connectivity among *Nyctalus lasiopterus* populations

The connectivity analysis among *N. lasiopterus* occurrence localities, based on the weighted suitability map obtained under current conditions, resulted in a dense, vast connection between northeastern Spain and southwestern France, overriding the Pyrenees (Fig. 3A). Moreover, this connectivity core is linked to other smaller corridors systems in southern France (Haut-Languedoc chain and coasts of southwestern Mediterranean France) and the Azahar Coast. On the other hand, smaller intra-connections emerged in Andalusia, Julian pre-Alps, and Bosphorus, even though there are few occurrence localities in the surroundings of the two latter. Also, the pre- and sub-Alpine Italian areas showed medium connectivity values, potentially linking the western systems to the eastern ones. On the contrary, the central European (Slovakian) occurrence cluster resulted in lower intra- and inter-connections.

When analysing future changes in connectivity through the SCCI, a loss can be observed in the southern part of the Iberian Peninsula–southwestern France corridor system, as well as in the corridors emerging in western Andalusia, Bosphorus and Slovakia (Fig. 3B). On the other hand, concurrent stability is predicted for the Haut-Languedoc chain and coasts of southwestern Mediterranean France. In contrast, connectivity increases in the northern part of the Julian pre-Alps system (Fig. 3B). Also, significant connectivity gains are observed for northern Portugal, southern Spain and northern France; in a similar way, central and eastern Europe gain connections across broad areas. All these mentioned trends are more pronounced as the time horizon for model projection is shifted farther in the future (*i.e.*, connectivity changes are greater in 2070 than in 2030); also, the SSP5.85-based predictions generally resulted in higher connectivity changes than those based on SSP3.70, especially in areas showing currently high connectivity.

## Connectivity towards old-growth forests, niche overlap, and protection status

About the connectivity inferred from the actual *N. lasiopterus* occurrence localities towards old-growth forests, we found that major corridors exist where old-growth forests are scarce. In fact, *Sabatini et al. (2018)* reported a high number, or even clusters, of old-growth forests in central Italy, the Balkans, and central-western Europe, while the major corridor systems emerging from our analyses are located in the northern Iberian Peninsula, southern France and Italian Alps (Fig. 4A). In addition, another small corridor

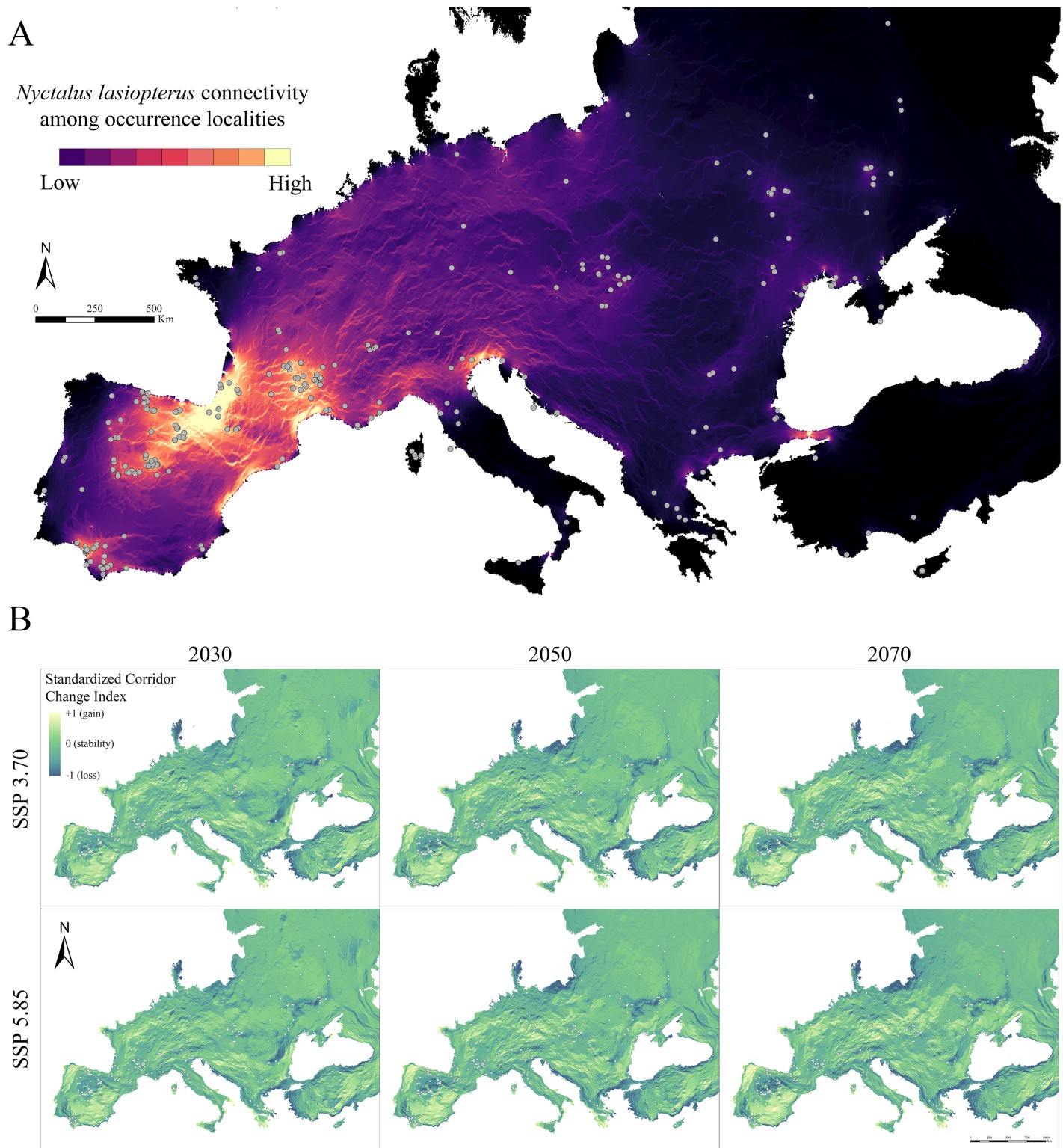

**Figure 3 Current and future inter-population connectivity.** (A) Pairwise landscape connectivity among *Nyctalus lasiopterus* occurrence localities, computed using the weighted suitability map obtained under current conditions as input in Circuitscape v. 5 software. (B) Standardized Connectivity Change Index (SCCI) showing the change in connectivity (blue-to-yellow color scale corresponding to loss-to-gain gradient) between each future scenario and current conditions.

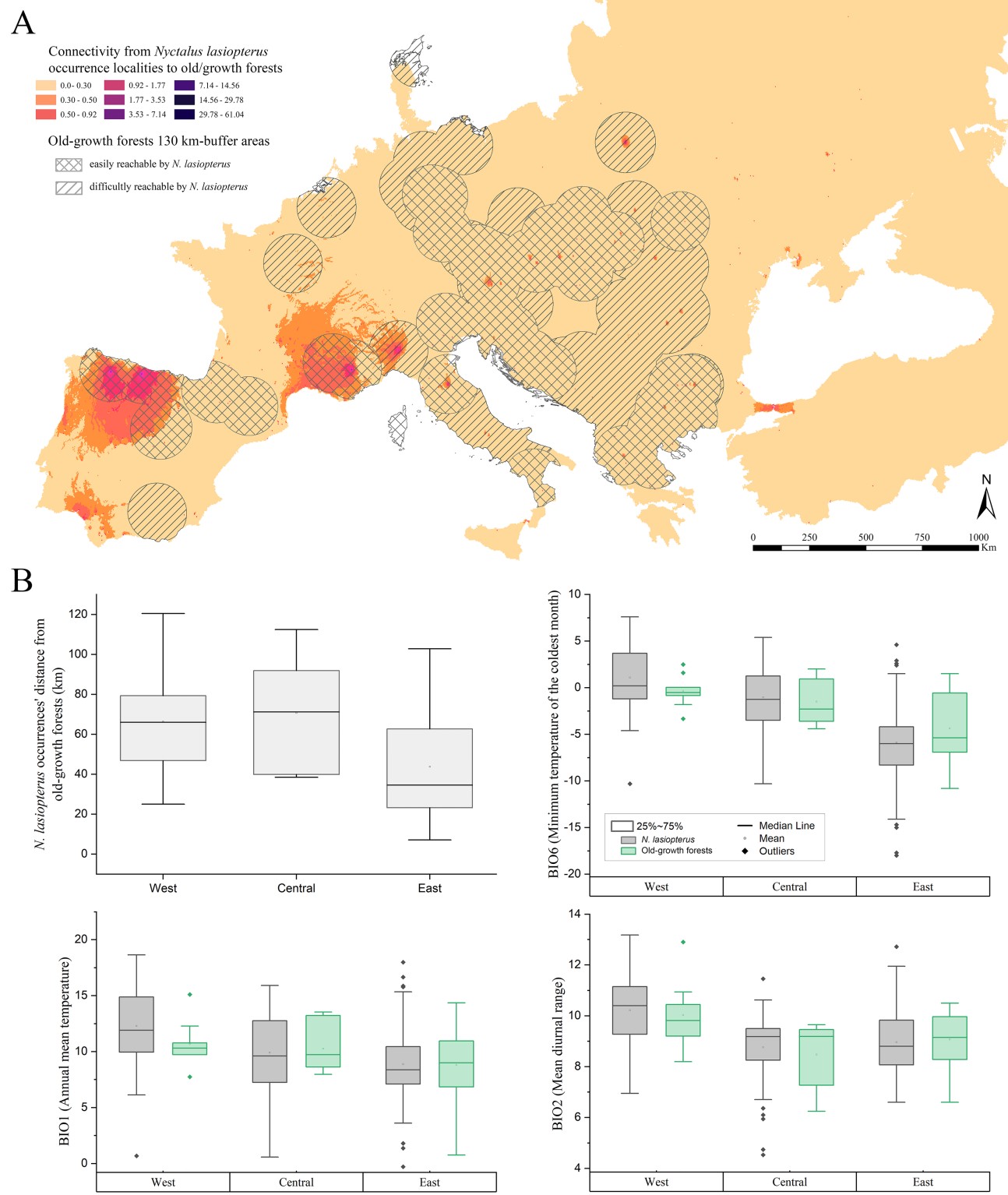

**Figure 4 Connectivity towards old-growth forests and bioclimatic characterization of occurrence localities.** (A) Landscape connectivity from current *Nyctalus lasiopterus* occurrence localities towards European old-growth forests. Crossed buffer areas indicate old-growth forests easily reachable by *N. lasiopterus* from its current occurrence localities, considering the species' known maximum daily dispersal distance (*i.e.*, ~130 km), while hatched buffer areas represent forests more difficult to reach. (B) For each of the three "longitude groups" of populations (*i.e.*, West, Central,

![PeerJ]

**Figure 4** (continued)
East) are shown, following clockwise direction from top-left panel: (i) distance between *N. lasiopterus* occurrence localities and old-growth forests; (ii) distribution of values for BIO6 in correspondence of the occurrence localities of *N. lasiopterus* (grey boxes) and old-growth forests (green boxes); (iii) distribution of values for BIO1 in correspondence of the occurrence localities of *N. lasiopterus* (grey boxes) and old-growth forests (green boxes); distribution of values for BIO2 in correspondence of the occurrence localities of *N. lasiopterus* (grey boxes) and old-growth forests (green boxes).

system is observed in Andalusia, which is more linked to the farther old-growth forests in the north than to the nearer ones, located eastward.

The SCCIs calculated on connectivity modelled between *N. lasiopterus* occurrences and old-growth forests located within the 130-km buffer show overall corridors' stability in the future except for the Pyrenees, where a slight increase in connectivity is predicted for all the scenarios (Fig. S3A), and for Andalusia, where instead a slight loss is predicted. Similarly, the areas outside the 130-km buffer show stable connectivity, except for the gains reported in the eastern Alps, northern France, and Denmark (Fig. S3B).

The three bioclimatic variables contributing the most to the climate-based ENM (*i.e.*, BIO6, BIO1 and BIO2) show decreasing values along the west-east longitude gradient, when sampled in *N. lasiopterus* presence localities. A similar trend emerges for the same values measured in old-growth forests' occurrences, altough these latter show higher variability (Fig. 4B). A mixed trend is instead observed for the distances between *N. lasiopterus* occurrences and old-growth forests, with the eastern ones being closer and the central and western ones occurring at more considerable distances (Fig. 4B).

In the bioclimatic space modelled through the 'hyperoverlap' R package, we found a separation (*i.e.*, non-overlap) between the western and eastern *N. lasiopterus* populations, with the central ones overlapping with both (Fig. 5A). When examining bioclimatic niche overlap between the distinct longitude groups of *N. lasiopterus* and old-growth forests, eastern *N. lasiopterus* populations overlap with all the forests' categories except the western ones, whereas the western *N. lasiopterus* populations overlap only with western forests (Fig. 5B). Central *N. lasiopterus* populations overlap with 'Old-Growth Forest' and 'Long Untouched Forest' categories from both the eastern and central longitude groups. Moreover, *N. lasiopterus* eastern populations are the only ones overlapping with the "near-virgin forest" category (Fig. 5B).

Finally, we found 59 *N. lasiopterus* occurrences (out of 260) falling into protected areas, of which 15 in Natura 2000 sites, eight in Nationally Designated Protected Areas, and 36 covered by both (*i.e.*, where these two kinds of protected areas overlap). The median distance between these *N. lasiopterus* occurrence localities and old-growth forests ranges from 20 km to nearly 120 km, depending on the considered longitude group and category of protected areas (Fig. 6).

## DISCUSSION

Many ecological aspects of *Nyctalus lasiopterus* are still poorly known, to the point of this species being defined "enigmatic" (*Naďo et al., 2019*), and distributional data are still accumulating (*Snit'ko & Snit'ko, 2021*; *Ibáñez & Juste, 2022*). In similar cases, recent developments in Ecological Niche Modelling permitted to individuate climate-related

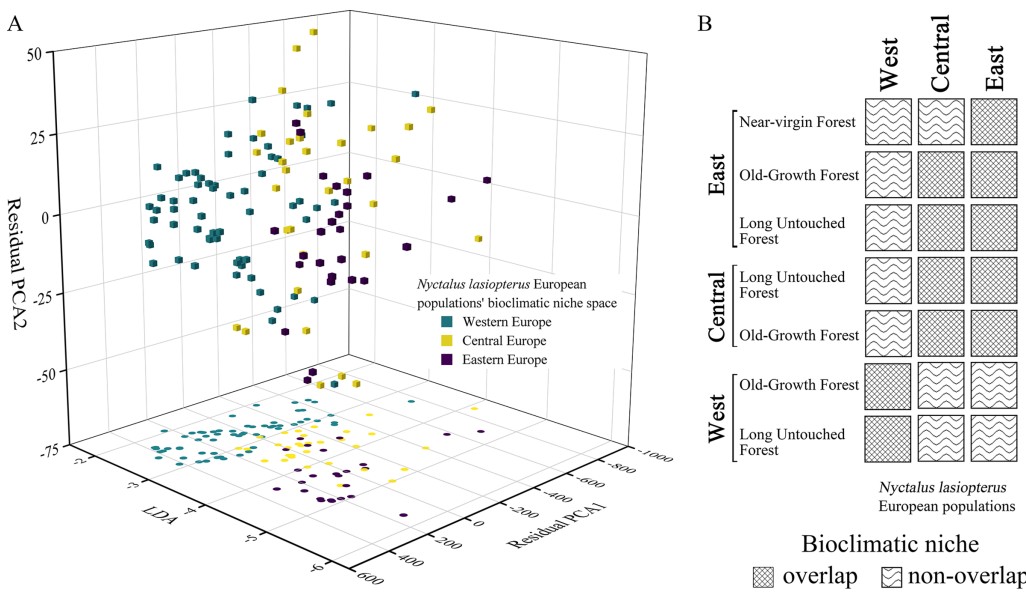

**Figure 5 Bioclimatic niche overlap among the three longitudinal groups of giant noctule populations and with old-growth forests.** (A) Distribution of *Nyctalus lasiopterus* occurrences in the bioclimatic niche space derived using the customized functions from 'hyperoverlap' R package, distinguishing among western (aquamarine), central (yellow) and eastern (violet) populations. (B) Bioclimatic niche overlap, for each "longitude group", among the different categories of European old-growth forests and *N. lasiopterus* populations.

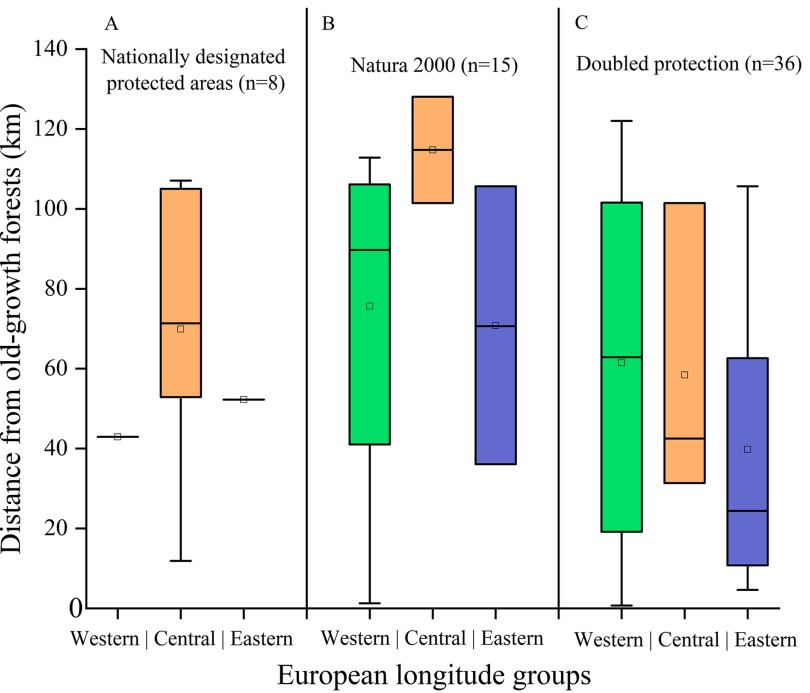

**Figure 6 Distance of protected *Nyctalus lasiopterus* occurrence localities from old-growth forests.** Distance from old-growth forests of the *Nyctalus lasiopterus* occurrences falling within Nationally Designated Protected Areas (A), Natura 2000 sites (B) or both these protected areas categories (C), for the three "longitude groups" of populations.

drivers influencing the distribution of many taxa of conservation interest (*Franklin, 2013*; *Cerasoli, Iannella & Biondi, 2019*; *Console et al., 2020*; *Gusmão et al., 2021*; *Sistri et al., 2022*), shedding light into hidden biogeographic (*Reino et al., 2017*; *Iannella, D'Alessandro & Biondi, 2018*; *Brunetti et al., 2019*) and ecological (*Mammola & Isaia, 2017*; *Di Musciano et al., 2020*; *Cerasoli, D'Alessandro & Biondi, 2022*) phenomena.

In this research, the climate-based ENMs indicated the minimum temperature of the coldest month (BIO6) as the most contributing variable for *N. lasiopterus*, likely driving the north-eastern shift of climatic suitability for this species under the considered future scenarios, a trend observed also for other taxa (*Iannella, D'Alessandro & Biondi, 2020*; *Guo et al., 2021*). Weighted suitability, resulting from merging climate-based predictions with topographic and habitat information, confirmed this trend: in fact, the areas currently showing the highest weighted suitability values are predicted to contract in the western portion of the species' range, balanced by an increase in medium-to-high suitability values eastwards. This is shown to affect landscape connectivity, which is in fact predicted to increase, in the future, among the central-eastern *N. lasiopterus* populations; contrarily, the easternmost populations would experience a loss of connectivity, thus posing them at a high risk of isolation, considering their actual low connectivity. Almost null connectivity emerged for the isolated occurrence localities from the Tuscan Archipelago and southern Italy, which have been recently hypothesized to represent doubtful records in the chapter of the new Handbook of the Mammals of Europe focusing on the giant noctle (*Ibáñez & Juste, 2022*).

The current high-connectivity asset in western Europe, clearly defined in specific areas of northern Spain, southern France and western Italian Alps, is also observed when modelling corridors between *N. lasiopterus* populations and old-growth forests.
If considering these connections and their possible accessibility, the forests in northern Spain and southern France show an evident importance in terms of conservation. On the contrary, the mild northward connection observed in southwestern Spain, where old-growth forests do not occur, breaks up in south-eastern Portugal. From a conservation perspective, this disruption in connectivity may be considered as less alarming than in other cases because previous research showed that *N. lasiopterus* populations occurring in Andalusia use holes within trees located in urban green, such as *Platanus* sp. and the palm *Washingtonia filifera*, for roosting, and are used to move several kilometers to find other suitable areas (*Popa-Lisseanu et al., 2008*; *Popa-Lisseanu, Bontadina & Ibáñez, 2009*). Instead, two main conservation concerns emerge more eastward. First, the northern Italian old-growth forests' corridor system could be exploited by *N. lasiopterus* not so easily, as no occurrences are found within the 130 km buffer. Secondly, the central-eastern systems of corridors host a number of old-growth forests but high connectivity emerges only in the neighbourhoods of these latter. The shorter distances among the eastern *N. lasiopterus* populations and old-growth forests may result from this lack of long-distance connections coupled with the climatic conditions characterizing eastern forests. The trend we have found suggests that forests take on great importance for *N. lasiopterus* when regional climatic conditions are suboptimal. In fact, in western Europe the distance of *N. lasiopterus* occurrences from old-growth forests is relatively high, despite vast corridors were predicted

there; at the same time, in this region the values of the selected climatic predictors fall into the intervals predicted as highly suitable for the species by the climate-based ENM. On the other hand, areas covered by the eastern old-growth forests generally showed a larger range of values for BIO6, BIO1 and BIO2 compared to the western ones; this resulted in higher overlap for the former group with climatic conditions characterizing *N. lasiopterus* occurrence sites. Thus, although current climatic suitability for *N. lasiopterus* at the regional scale is lower in the East, eastern old-growth forests appear to host suitable conditions for the giant noctule to thrive. Indeed, micro-climatic conditions or other factors (*e.g.*, micro-habitat availability, specific trophic resources, *etc.*) could allow populations of some bat species to persist even within an environmental matrix being unfavorable at a meso- or macro-scale (*Popa-Lisseanu, Bontadina & Ibáñez, 2009*; *Ancillotto et al., 2015*).

The niche overlap tests confirm the strong link between *N. lasiopterus* populations from the distinct "longitude groups" and the corresponding forests, highlighting a clear west-*vs*-east pattern. Central European groups overlap with the eastern ones, while the western populations form a separate group; also, eastern near-virgin forests overlap with eastern *N. lasiopterus* populations.

These outcomes should inform the protection strategies for both the giant noctule and the habitats it currently dwells in and/or which could possibly be used in the future. In fact, despite its alarming conservation status and scarce distributional and demographic information, *N. lasiopterus* is scarcely protected at present (*Alcaldé, Juste & Paunović, 2016*), which is confirmed by our finding of only 59 occurrence localities being covered by protected areas. The giant noctule is listed in Eurobats and Bern conventions, which however do not apply across the entire species' range, so that the degree of protection varies among the distinct countries; at the continental scale, it is listed only in the Annex IV of the European Habitats Directive 92/43/EEC. The fact that protected areas heterogeneously cover the territories linking *N. lasiopterus* occurrences and old-growth forests, coupled with evidence of distinct environment-occurrence relationships of western populations compared to the central and eastern ones, suggests that distinct management practices are needed along the European longitudinal gradient. Evidently, the co-occurrence of Nationally Designated Protected Areas and Natura 2000 sites permits the highest protection, but the total number of *N. lasiopterus* localities occurring within territories covered by this "doubled" protection is scarce. The western occurrence localities and their potential corridors towards old-growth forests resulted scarcely protected, despite landscape connectivity being noticeably high in this region. Thus, specific protection measures should be applied to northern Spain and southwestern France. As an exception, the Andalusian populations, occurring in urbanized environments (*Popa-Lisseanu, Bontadina & Ibáñez, 2009*), seem to have adapted to the non-optimal habitat thanks to the suitable climate (the most suitable of the entire study area), also considering that no large and/or mature forests exploitable as roost sites are available in the neighboring areas. Therefore, the best management choice in the western part of the study area would be to avoid disturbances in the roosting Andalusian sites while keeping, and possibly expand, the current Natura 2000 sites in northern Iberia and southern France.

On the other hand, central and eastern European areas presently host suboptimal climatic conditions, but a higher proportion of *N. lasiopterus* occurrences located near old-growth forests, partially covered by a doubled protection. We would like to also underline that some specific actions should be undertaken in areas where alien species could compete and cause disturbance to the giant noctule. For instance, the Seville population seems to be more and more threatened by the invasive alien rose-ringed parakeets (*Psittacula krameri*) (*Hernández-Brito et al., 2018*). This species, which was also recently found to damage some common noctule (*N. noctula*) populations in Italy (*Giuntini et al., 2022*), competes for tree cavities with the giant noctules, causing the disruption of colonies, injures or even kills of individuals (*Hernández-Brito et al., 2018*). Finally, the predicted northeastern expansion of climatic suitability for the giant noctule and the exclusive niche overlap between its central-eastern populations and the corresponding old-growth forests suggest that well-targeted conservation measures for old-growth forests in these territories coupled with accurate monitoring of *N. lasiopterus*, in terms of both possible future range expansion and local population dynamics, should be the priority for the next years.

### Funding

This research was supported by Ministero dell'Istruzione, dell'Università e della Ricerca (A.I.M. Project PON R&I 2014–2020) through the grant No. 1870582 awarded to Dr. Mattia Iannella. The funders had no role in study design, data collection and analysis, decision to publish, or preparation of the manuscript.

### Grant Disclosures

The following grant information was disclosed by the authors:
Ministero dell'Istruzione, dell'Università e della Ricerca: 1870582.

### Competing Interests

The authors declare that they have no competing interests.

### Author Contributions

- Mattia Iannella conceived and designed the experiments, performed the experiments, analyzed the data, prepared figures and/or tables, authored or reviewed drafts of the article, and approved the final draft.
- Urbana Masciulli conceived and designed the experiments, authored or reviewed drafts of the article, and approved the final draft.
- Francesco Cerasoli performed the experiments, authored or reviewed drafts of the article, and approved the final draft.
- Michele Di Musciano analyzed the data, prepared figures and/or tables, authored or reviewed drafts of the article, and approved the final draft.
- Maurizio Biondi conceived and designed the experiments, performed the experiments, authored or reviewed drafts of the article, and approved the final draft.

## Data Availability

The raw data and code are available in the Supplemental Files.

## Supplemental Information

Supplemental information for this article can be found online at http://dx.doi.org/10.7717/peerj.14446#supplemental-information.

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
