# Peer review of "Assessing future shifts in habitat suitability and connectivity to old-growth forests to support the conservation of the endangered giant noctule"

_PeerJ, doi:10.7717/peerj.14446_

## Round 0.1 · original submission · Major Revisions

This manuscript has scientific merit, but there are some major issues; hence it is recommended for major revision as suggested by both reviewers.

·

Basic reporting

no comment

Experimental design

no comment

Validity of the findings

no comment

Additional comments

I found the manuscript presented extremely well done, both in its formal presentation, clarity, and scientific soundness.
I only have two very minor comments:

1. the authors present a map of the retrieved presence records; I found some mismatches between this map and the most recent reviewed one from the new Handbook of Mammals of Europe (https://link.springer.com/referenceworkentry/10.1007/978-3-319-65038-8_65-1#Sec4), and I have the suspect that (at least very few) locations may actually represent old or unconfirmed records that the authors aimed at filtering. For example, Ibanez reports records from Sicily and the Tuscan Archipelago as old/isolated/unconfirmed. Were these data excluded from the modeling by the authors? I don't think these fewest points actually seriously influenced the models, yet this may be a factor worth considering.

2. The conservation considerations by the authors are all very sound and well supported; I suggest though to also include other biological interactions besides that may be affected by climate change (i.e., besides predator-prey mismatches between the noctule and insects/passerine birds), such as competition with alien species (https://doi.org/10.1098/rsos.172477)

Reviewer 2 ·

Basic reporting

Dear Authors,
the subject of your study is relevant and of current interest considering the urgent need to assess the effects of climate change on biodiversity. The manuscript is quite well written, as well as the raw data provided. However, the paper is not yet ready for publication and I think there are a few modifications that would enrich it. I will explain my major comments in the relative sections below.
The paper structure is in accordance with the journal guidelines, however, the reference list has a different font from the text of the manuscript. Same for the figures’ captions that look larger.
Concerning the background, the authors should provide more information on the European populations of N. lasiopterus, as well as the distribution/status of the old-growth forests in the study area. In particular, I was wondering if other factors of isolation (e.g., genetics, etc..) between European populations have been investigated in the literature.

Experimental design

The research question is clear and fits the knowledge gap on the broad-scale and connectivity model of this species. However, the methods are not clear and need further explanations. In particular:

Study area and spatial data

L105 occurrence dataset: concerning the ‘literature data’ you need to specify how you set up the literature search, what references you chose, and how you obtained these data (did you request the data to the authors or did you digitize the locations?). Moreover, how many occurrences did you get from both sources (literature and GBIF) before the cleaning process?
L110 ‘excluded..coarse geographic coordinates’: this sentence is ambiguous and you need to quantify this ‘coarse’ resolution. Moreover, when citizen science data are used (e.g., GBIF), an exploration and correction of sampling bias are usually required because these data are affected by multiple sampling biases (Hortal et al., 2015 Annu. Rev. Ecol. Evol. Syst.; Cosentino & Maiorano, 2021 Ecol. Inf.). No sampling bias analysis is present in the manuscript and should be integrated to reduce potential overfitting and spatial dependence between the occurrences.
L128-130: the description of the river dataset and its processing is scarce and more details are needed (e.g., what does ‘river order’ stand for?). Moreover, it is not clear the final resolution of both river and elevation datasets to be implemented in the couple-weigh framework.
L125-141: the structure of this paragraph is confusing because it leads the reader to think that you will use all these variables to calibrate the ENM model. I suggest specifying for what and in which step of the framework each variable will be used.
L137-141: For greater clarity with the following methods, you should specify what is the resolution and the total area occupied by the old-growth forests after the layer processing.

Ecological Niche Modelling

L144: You need to motivate the choice of the ‘gbm’ algorithm among the other machine learning algorithms.
L145-146: VIF results should be supported by a table.
L150: the strategy for the selection of pseudoabsences should be deeply argued with a reasonable rationale. Moreover, what is the impact of this background selection on the connectivity analysis?
L144 and L169: In general, both the calibration and the projection areas are not clear over the manuscript as well as the spatial resolution of the ecological niche model.

Post-modelling analysis

Generally speaking, you should give more technical information about these analyses, rather than giving the names of the tools used. In particular:
L193-195: The raw data for the river dataset is not clear. How water is represented (e.g., % of water in a cell, distance to water..)?
Landscape connectivity
L221: concerning the 130 km buffer around old-growth forests, it is not totally clear to which analyses it was applied.

Spatial and statistical analyses

L226-240: this paragraph is not clear. You need to explain better both the rationale and the analytical framework.
L241: the validity of this method to assess the protection status should be deeply argued and supported by references.
L248: Since you mentioned several R packages, I guess you used R too..is that a typo?

Validity of the findings

Results

L259-262: These results should be supported by a table showing the variable importance of all predictors. Moreover, you need to explain how did you measure the variable importance in the methods section.
L270: This sentence is too vague. You need to specify a suitability threshold that defines ‘highly suitable areas’.
L279-285:The future projection results are given for each time frame and SSP, but the three General Circulation Models (GCMs) used are not mentioned. Did you calculate the average between the GCMs to summarize the results? You need to clarify this in the methods section.
In general, the results section should be improved by using more consistent terminology to make the reading clearer. For example, to refer to the different areas of your study area you used different terms such as latitude-longitude range (e.g., L271), specific countries (e.g., L273-274), and macro areas (e.g., ‘western areas’, ‘north-eastern areas’) which makes the text hard to follow. Be consistent also in referring to tables and figures (e.g., L275 ‘Figure 1’, L279 ‘Fig. 2A, B’, L285 ‘Fig. 2 C,D’).

The discussion is well structured, however, I suggest using simple and ordered sentences, and avoiding sentences nested in another sentence (e.g., L394-397) which make the text chaotic. Then, I have two specific comments:
L385: Could you provide evidence that the species is not located in these areas because it is actually absent (= not found) in the 130 km buffer, and not because of a sampling bias (e.g., areas never sampled for this species)?
L397-399: for this outcome, did you consider the potential implications of micro-climatic factors?

Additional comments

Moreover, here few typos:

L53: Nyctalus lasiopterus (Schreber, 1780)
L393: “fr” in “for”
Figure 2 C and D: it is not clear which part of the study area is represented; moreover in the figure caption is mentioned the ‘Interpolation’ but from the methods section any interpolation analysis is not mentioned.

Finally, with the methods thus described the paper cannot be accepted for publication and requires deep editing.

---

## Round 0.2 · accepted · Accept

The authors have incorporated the suggestions made by the reviewers. Hence, my recommendation is to accept it.